# Flexibility in choosing decision policies in gathering discrete evidence over time

**Masoumeh Golmohamadian[1], Mehrbod Faraji[1,2], Fatemeh Fallah[1], Fatemeh Sharifizadeh[1], Reza Ebrahimpour[3]***

1 School of Cognitive Sciences (SCS), Institute for Research in Fundamental Science (IPM), Tehran, Iran, 2 Department of Computer Engineering, Shahid Rajaee Teacher Training University, Tehran, Iran, 3 Center for Cognitive Science, Institute for Convergence Science and Technology (ICST), Sharif University of Technology, Tehran, Iran

* ebrahimpour@sharif.edu

## Abstract

The brain can remarkably adapt its decision-making process to suit the dynamic environment and diverse aims and demands. The brain's flexibility can be classified into three categories: flexibility in choosing solutions, decision policies, and actions. We employ two experiments to explore flexibility in decision policy: a visual object categorization task and an auditory object categorization task. Both tasks required participants to accumulate discrete evidence over time, with the only difference being the sensory state of the stimuli. We aim to investigate how the brain demonstrates flexibility in selecting decision policies in different sensory contexts when the solution and action remain the same. Our results indicate that the decision policy of the brain in integrating information is independent of inter-pulse interval across these two tasks. However, the decision policy based on how the brain ranks the first and second pulse of evidence changes flexibly. We show that the sequence of pulses does not affect the choice accuracy in the auditory mode. However, in the visual mode, the first pulse had the larger leverage on decisions. Our research underscores the importance of incorporating diverse contexts to improve our understanding of the brain's flexibility in real-world decision-making.

## Introduction

Imagine you are walking in a meadow with long grass. Suddenly, you see something in the grass, but it disappears when you look closer. Later, you see it again briefly and have an opportunity to gather more information about what you saw before. You need to identify what you are seeing, so you can escape or defend yourself if it is a predator like a lion. The dense and tangled grass and the color similarity between flowers, grass, and what you see make it hard to identify. The time interval between your two sightings also may influence your decision.

Now imagine yourself in the same meadow again, but this time you do not see any strange thing in the grass. Instead, you hear a short and unfamiliar sound that differs from the natural sounds of the meadow. Then everything becomes quiet again. After a while, you hear a similar sound again briefly. You need to determine whether the sound comes from a wild animal

**Data Availability Statement:** The dataset employed in this study is available to the public and is cited below: Golmohamadian, Masoumeh; Faraji, Mehrbod; Fallah, Fatemeh; Sharifizadeh, Fatemeh; Ebrahimpour, Reza (2024), "Flexibility in Choosing

Decision Policies in Gathering Discrete Evidence Over Time", Mendeley Data, V1, DOI: 10.17632/78sht3jsr2.1.

**Funding:** The author(s) received no specific funding for this work.

**Competing interests:** The authors have declared that no competing interests exist.

hidden in the grass. The sounds of wind, the rustling of plants, and your footsteps create background noise that interferes with the auditory information you receive and makes it difficult for you to recognize what you are hearing and prepare yourself for it. Your decision may also be influenced by the time interval between your two hearings.

How much information do you need to make such a decision in the previous situations? You may be harmed if you do not decide quickly enough. But the auditory and visual data are vague and you may not want to react hastily. In realistic environments, for making perceptual decisions, decision-makers need to first identify the context and set up the decision-making process accordingly [1–3]. They need to choose sensory information, actions, payoffs, solutions, and policies that are relevant to the decision. This decision-making structure has a hierarchy and allows them to flexibly adapt their decisions to generate various options to satisfy their needs [4].

In the literature, flexibility in decision making, often referred to as context dependency, reflects the brain's ability to adjust its behavior and neural activity across different tasks to meet varying demands. This adaptability is organized into three levels, each representing distinct types of flexibility that enhance decision making performance [4]. At the highest level, decision-makers can adopt different solutions for a task, such as integrating, differentiating, or even disregarding sensory inputs depending on the context. For example, evidence accumulation is the optimal solution for tasks involving the discrimination or categorization of stable sensory information [5, 6]. However, other task conditions may call for different approaches. When comparing the magnitudes of two stimuli, subtracting inputs is more effective than integrating them [7, 8], whereas, for detection tasks against a stable background, differentiation is a better strategy than integration. At the next level, decision-makers adjust decision policies by modifying parameters within decision-making models to achieve optimal outcomes. These adjustments support flexible computation, enabling decision-makers to weigh sensory inputs, prioritize evidence, assess costs and rewards [9], and manage urgency to respond [10]. Finally, perceptual decisions involve mapping sensory stimuli to actions, but existing decision-making models typically assume a fixed stimulus-action association and focus on evidence integration rather than flexible stimulus-action mapping [11–13]. Together, these levels of flexibility demonstrate that decision-making is not simply about processing information but involves a multi-layered, adaptive system capable of dynamically adjusting solutions and policies in response to the immediate context, thereby enabling more effective navigation of complex and real situations.

To study this flexibility, the field of perceptual decision-making has undergone some important changes [14]. One of these transformations is the use of naturalistic stimuli instead of simplified sensory stimuli that vary along a single dimension, such as the direction of moving dots changing from left to right [15, 16]. This approach aims to examine the decisions that humans make daily based on the stimuli that they experience in the real world [17, 18].

Naturalistic stimuli often have multimodal features such as shape, texture, and sound, each of which can influence human perceptual decisions. Most research on perceptual decision-making has focused on unimodal contexts. Multimodal integration is also essential for understanding how decisions are made in the real world, and this is another approach that recent studies have explored [13, 18]. In this approach, researchers examine how the brain integrates multisensory information when evidence from different modalities is presented to a participant simultaneously [18–21]. Specifically, they demonstrated that the synergy between sensory modalities results in a faster accumulation of sensory evidence and more accurate perceptual decisions [18, 20, 22, 23]. However, perceptual decision-making regarding naturalistic stimuli could be studied in another way that examines how the brain processes sensory information

from different modalities associated with the same stimulus individually and compares similarities and differences among them [24, 25].

The brain's proficiency in categorizing objects regardless of sensory modality is evidence of how the brain handles visual and auditory stimuli separately yet effectively. The ventral visual pathway is a functional stream involved in the visual recognition of objects [26, 27]. However, neural associations linked to auditory categorization have been discovered in diverse brain areas of the ventral auditory pathway [28, 29]. Despite the differences in the pathways engaged in processing visual and auditory information, the brain exhibits remarkable proficiency in categorization, irrespective of the sensory modality through which the information is perceived [30]. So, identifying the differences in the behavioral and computational mechanisms of the brain in making decisions based on these two sensory modalities seems to be essential.

In this study, similar to the above example, we explore perceptual decision-making when a subject gathers evidence from distinct cues over time in separate auditory and visual experiments. We employ well-established visual and auditory face-vs-car categorization tasks. Both tasks have the same structure, but the sensory modality of stimuli is different. We choose cars and faces as naturalistic stimuli [31] that have multimodal features, shape and sound, to follow recent research approaches in perceptual decision studies. Our goal is to examine the brain's flexibility in choosing decision policies in different sensory contexts when it has taken the same action and solution.

For further comparison, we compare our findings to the results of previous research [32] that conducted a similar experiment but on a motion direction discrimination task. Its findings indicate that subjects can effectively integrate data from two visual motion cues to increase their accuracy and the time interval between the cues did not affect the performance [32, 33]. Finally, the second cue had a greater impact on the decisions made by the participants [32, 34]. By comparing the decision-making process in our visual and auditory tasks with the earlier task, we show that the brain can use similar or different decision policies to make decisions that have similar solutions and fixed actions. This demonstrates the brain's flexibility in changing decision policies in different contexts.

Another aspect of our research involves investigating the confidence of participants when performing visual and auditory tasks. In this approach, we seek to gain more insight into how the sense of assurance in subjects' decisions is affected by different sensory modalities. In several studies, the subjective sense of confidence [35] as a metacognitive factor [36, 37] was examined along with various perceptual decision-making tasks [38–43]. Here, our results demonstrate that participants' confidence increases or decreases with their choice accuracy, regardless of the sensory modality of the stimuli.

## Materials and methods

In this study, we conducted two separate experiments: a visual experiment and an auditory experiment. Both experiments were identical, except for the stimuli used, which were presented through different senses (visual and auditory).

### Participants

In this study, 15 adult human subjects (9 male and 6 female) with a mean age of 27.4 and STD of 6.04 participated in both visual and auditory experiments. All subjects, except for two of the authors (M.G and M.F), were unaware of the experiment's purpose. They also had normal or corrected-to-normal vision and hearing and provided written informed consent before their participation. The experimental procedure was approved by the ethics committee at the Iran University of Medical Sciences. The project was found to be in accordance to the ethical

principles and the national norms and standards for conducting Medical Research in Iran. The approval ID is IR.IUMS.REC.1399.1282. Written consent was obtained from all participants. The recruitment period for this study started on 20/11/2021 and ended on 15/12/2023.

## Visual stimuli

A set of 24 grayscale images was utilized in this study, with 12 images of faces and 12 of cars (image size 500 × 500 pixels, 8 bits per pixel) comprising this set. These images are adapted from the study by Franzen et al. [18] and are all equalized in spatial frequency, luminance, and contrast and have identical magnitude spectra (mean magnitude spectrum of all images). The phase spectra of the images were manipulated to control the task difficulty, employing the weighted mean phase (WMP) approach from Dakin et al. [31, 44]. The phase spectrum of each image was blended with the phase of uniform random noise, according to the WMP method, to generate stimuli with six distinct levels of strength (20%, 25%, 30%, 35%, 40%, 45%), with the stimulus strengths of 0.00 and 1.00 equal to pure noise and pure image, respectively. All stimuli were displayed 50 ms on a gray background (RGB: 0.5, 0.5, 0.5) using the MATLAB Psychophysics Toolbox Version 3 [45, 46].

## Auditory stimuli

Similar to the visual mode, we processed 12 face- and 12 car-related sounds to create the face and car auditory stimuli, human speech and car/street sounds, respectively. All sounds were obtained from freely available online sources and were resampled and cut to have a constant sampling rate of 44.10 kHz and a length of 50 ms. We multiplied 10 ms of the sounds' onset/ offset with ascending/descending cosine ramps to reduce the effect of sudden changes in the stimuli and subsequently normalized all sounds by their standard deviation, followed by an amplitude reduction of 25%. The resulting sounds were then embedded in Gaussian white noise, with six different levels of noise strength (2.5, 2.0, 1.5, 1.0, 0.5, 0.0), to generate stimuli with six levels of relative noise-to-signal ratio [18]. In other words, we presented sounds to participants using six different levels of relative noise-to-signal ratios (250%, 200%, 150%, 100%, 50%, 0% of added noise). The final stimuli were presented binaurally through headphones (Sony MDR-ZX310) for 50 ms using the PsychPortAudio library from Psychophysics Toolbox Version 3 [45, 46] to allow for precise control over the temporal gaps in the main experiment.

Both experiments were controlled by a 64-bit-based machine (32 GB RAM) with 11th Gen Intel(R) Core (TM) i7-11700 @ 2.50GHz processor, running Windows Professional 7 and Psychophysics Toolbox Version 3. All images were presented on an LG monitor (20MP38AB, resolution; 1440 × 900 pixels; refresh rate set to 60 Hz). The participants were seated 75 cm from the stimulus display and saw each image as ~11 × 11 degrees of visual angle.

## Visual and auditory object categorization tasks

The behavioral tasks in this study have been adapted from the works of Kiani et al., Franzen et al., and Philiastides & Sajda [18, 31, 32]. More precisely, our study involved two separate experiments: a visual experiment and an auditory experiment. Both experiments were similar, except for the stimuli being presented through different senses (Fig 1A and 1B). The subjects were seated in an adjustable chair in a semi-dark room and asked to categorize the stimuli presented to them as either faces or cars in each trial of each task. In the visual object categorization task, the stimuli were noisy images, while in the auditory object categorization task, the stimuli were noisy sounds presented in one or two pulses (Fig 1A and 1B).

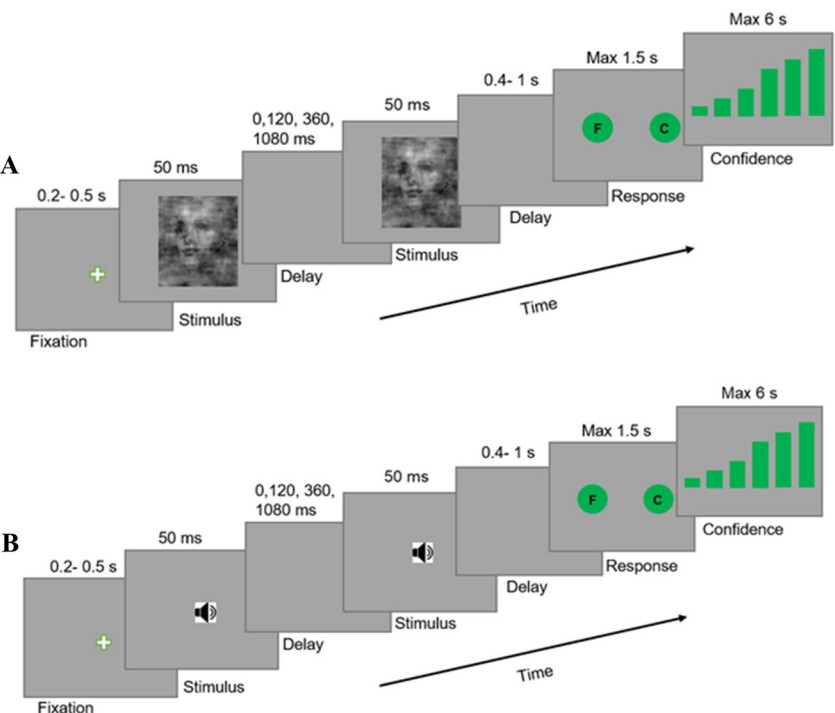

**Fig 1. Participants categorize noisy images or sounds in a double-pulse object categorization task presented in visual and auditory modalities.** Object categorization task with two brief pulses of information in two separate sensory modes: visual and auditory. In visual mode (A), the stimuli were images and in auditory mode (B), the stimuli were sounds. In each mode, subjects were asked to categorize noisy representations of faces and cars, presented in one or two 50 ms pulses and then report their confidence. The temporal gap between pulses ranged from 0 to 1.08 s. In both sensory modes, the strength of stimulus, image or sound, was randomized across trials and between two pulses, but the category of object (face or car) was the same for both pulses on double-pulse trials.

The experimental paradigm, in both visual and auditory tasks, was divided into two phases: the training phase and the testing phase. In the training phase of the visual image discrimination task (face-vs-car), subjects completed four blocks of 120 single-pulse trials. We utilized five levels of visual evidence for each image (20%, 25%, 30%, 35%, and 40%). Throughout the entire training session, these trials were equally distributed among the categories of stimuli and strength levels, resulting in 48 trials for each of the 10 possible trial types. Notably, in each block, all face images and car images across all strength levels were presented once.

During the training phase of the auditory sound discrimination task (face/speech vs car/ street sounds), we presented sounds to participants using six levels of relative noise-to-signal ratios (250%, 200%, 150%, 100%, 50%, and 0% of added noise). In the auditory training phase, subjects completed six blocks of 96 single-pulse trials. Similar to the visual task, these trials were equally divided among the s timuli category and strength levels during the entire training session, yielding 48 trials for each of the 12 possible trial types.

Overall, we presented 480 trials for the visual discrimination training task split into four blocks of 120 trials and 576 trials for auditory discrimination training tasks split into six blocks of 96 trials.

In both sensory tasks, during the training phase, a trial was initiated when the subject pressed the space button. This action was followed by the appearance of a white fixation cross (0.4° diameter) at the screen's center, which lasted for a short delay (200–500 ms,

truncated exponential distribution). Subsequently, the stimulus was presented for a fixed duration of 50 ms. For the visual task, the stimulus was displayed at the screen's center, while for the auditory task, it was presented via an empty grey screen and binaurally through headphones. Upon the completion of the stimulus presentation, a 400–1000 ms delay period (truncated exponential) was imposed through a grey empty screen. This was followed by a response screen consisting of two circles labelled with the letters 'F' for face and 'C' for car, which appeared on the left and right sides of the screen, respectively. The subject was required to report their decision by clicking on one of the circles within 1.5 seconds using a mouse. Following the declaration of their decision, six bars of varying heights appeared at the screen's center. These bars represented the subject's decision confidence, with the shortest bar on the left indicating the lowest level of confidence and the tallest bar on the right indicating the highest level of confidence. The subject was then required to report their confidence level by clicking on one of the bars with a mouse. Then, the subjects received visual feedback indicating the accuracy of their decision. This feedback was displayed as text at the screen's center, reading 'Correct' in green for correct decisions, 'Incorrect' in red for wrong decisions, and 'Too Long' in blue for time-out decisions. To initiate the next trial, the subject pressed the Space button.

During the testing phase, we applied six levels of stimulus strength (20%, 25%, 30%, 35%, 40% and 45%) in the visual object categorization task and six levels of relative noise-to-signal ratios (250%, 200%, 150%, 100%, 50%, and 0% of added noise) in the auditory object categorization task. In both tasks, subjects completed four blocks of 168 double and single-pulse trials. In almost 80% of all trials, the stimuli were presented to the subject in two discrete pulses, with the pulses being separated by a temporal gap or an inter-pulse interval (IPI). The duration of each IPI was chosen from a set of four values (0, 120, 360, 480, 1080 ms). In both tasks, the object was always consistent for the double-pulse trials, and subjects were informed of this fact. Three stimulus strength levels were employed for each pulse of the double-pulse trials, 25%, 30% and 35% for the visual task and 200%, 150% and 100% noise-to-signal ratios for the auditory task. These levels were selected based on subjects' performance in the training phase, as we required stimulus strength levels where decision categories are not easily distinguishable. In single and double-pulse trials in both tasks, each stimulus pulse was presented for a fixed duration of 50 ms. Double-pulse trials were divided equally among all possible trial conditions: a combination of decision category, IPI duration, and stimulus level. More specifically, there are 42 possible trial types, 6 single-pulse and 9×4 double-pulse trials. Subjects reported their decision and confidence similarly to the training phase and were instructed to perform as accurately as possible. It is important to note that we didn't provide any feedback to the participants throughout the testing phase (Fig 1A and 1B).

## Analysis of behavioural data

We employ different logistic regression models to evaluate the influences of visual/auditory stimulus parameters on binary responses. More precisely, we use a generalized linear model (GLM) which is a binomial error mode for fitting under maximum likelihood estimation. All models' input data were normalized. We also consider $Logit\,[p]$ as shorthand for $log\left(\frac{p}{1-p}\right)$ and $\beta_i$ as fitted coefficients.

The probability of a correct choice of visual and auditory *(V/A)* single-pulse trials is approximated by the following regression model:

$$Logit[P_{correct}] = \beta_0 + \beta_1 C \tag{1}$$

$$Logit[P_{correct}] = \begin{cases} \beta_0 + \beta_1 C^V : \text{for visual trials} \\ \beta_0 + \beta_1 C^A : \text{for auditory trials} \end{cases}$$

where $C^V/C^A$ is the image or audio coherence (strength) of the associated stimulus.

The effect of inter-pulse interval on the accuracy of visual or auditory double-pulse trials is evaluated by fitting the following regression analysis:

$$Logit[P_{correct}] = \beta_0 + \beta_1 C_1 + \beta_2 C_2 + \beta_3 T + \beta_4 C_1 T + \beta_5 C_2 T \qquad (2)$$

$$Logit[P_{correct}] = \begin{cases} \beta_0 + \beta_1 C_1^V + \beta_2 C_2^V + \beta_3 T + \beta_4 C_1^V T + \beta_5 C_2^V T : \text{for visual trials} \\ \beta_0 + \beta_1 C_1^A + \beta_2 C_2^A + \beta_3 T + \beta_4 C_1^A T + \beta_5 C_2^A T : \text{for auditory trials} \end{cases}$$

Where $T$ is the inter-pulse interval, and $C_1^V/C_1^A$ and $C_2^V/C_2^A$ are the coherence of the first and second visual/auditory pulses, respectively. Moreover, when the above equation is applied to double pulse trials with equal pulse strength ($C_1 = C_2$), the redundant regression terms ($\beta_2 C_2$ and $\beta_5 C_2 T$) were omitted. The null hypothesis states the accuracy is not dependent on the interval inter-pulse ($H_0: \beta_{3-5} = 0$).

To assess the effect of pulse sequence on the accuracy of visual or auditory double pulse trials, we fit the following regression model:

$$Logit[P_{correct}] = \beta_0 + \beta_1 [C_2 + C_1] + \beta_2 [C_2 - C_1] \qquad (3)$$

$$Logit[P_{correct}] = \begin{cases} \beta_0 + \beta_1 (C_2^V + C_1^V) + \beta_2 (C_2^V - C_1^V) : \text{for visual trials} \\ \beta_0 + \beta_1 (C_2^A + C_1^A) + \beta_2 (C_2^A - C_1^A) : \text{for auditory trials} \end{cases}$$

Where $C_1^V/C_1^A$ and $C_2^V/C_2^A$ are the coherence of the first and second visual/auditory pulses, respectively and $\beta_2$ represents the change in probability correct from the weak-strong pulse ($C_1 < C_2$) to the strong-weak one ($C_2 > C_1$) and $H_0: \beta_2 = 0$.

In order to examine whether there was an interaction between the two pulses, such as if a stronger pulse 1 would diminish the impact of pulse 2, we employed the subsequent regression:

$$Logit[P_{correct}] = \beta_0 + \beta_1 C_1 + \beta_2 C_2 + \beta_3 C_1 C_2 \qquad (4)$$

$$Logit[P_{correct}] = \begin{cases} \beta_0 + \beta_1 C_1^V + \beta_2 C_2^V + \beta_3 C_1^V C_2^V : \text{for visual trials} \\ \beta_0 + \beta_1 C_1^A + \beta_2 C_2^A + \beta_3 C_1^A C_2^A \text{ for auditory trials} \end{cases}$$

We use the regression analysis on every double-pulse trials. The null hypothesis is that the improved efficacy of the first pulse was related to increased sensitivity and not an interaction of pulses ($H_0: \beta_3 = 0$). It is important to note that $\beta_1 > \beta_2$ denotes a higher sensitivity to the first pulse on the decision-making.

## The *d'* sensitivity measure

The *d'* (d-prime) sensitivity measure [47] is a performance indicator that merges both the hit rate (*H*), which is the percentage of face images/sounds correctly classified by the observer, and the false alarm rate (*F*), which is the percentage of car images/sounds misclassified by the

observer, into a single standardized score. The formula for the d-prime measure is

$$d' = Z(H) - Z(F) \tag{5}$$

with $Z$ being the inverse of the normal distribution function. A higher absolute value of $d'$ indicates greater sensitivity to distinguishing between face and car, while values near zero suggest chance performance.

## Psychometric function

We use psychometric curves to describe the behavioural performance of our subjects. We plot the probability of correct against the stimulus strength. To determine the best-fitting cumulative Weibull distribution [48] for each data set, we employ a maximum-likelihood method [49]:

$$p = 1 - 0.5e^{-\left(\frac{s}{\alpha}\right)^{\beta}} \tag{6}$$

where $p$ represents probability correct, computed as a function of the stimulus strength $s$ for our stimuli. The parameters $\alpha$ and $\beta$ are fitted values that show the threshold performance (82% correct) and the slope of the curve, respectively.

## Statistical tests

Accuracy was defined as the ratio of correct responses to the total number of trials, expressed as a percentage. To compare double-pulse trials with different temporal gaps between the same and different stimuli strengths, we conducted a one-way ANOVA test. However, no significant difference was observed, leading us to forego post-hoc analysis. The Lilliefors test was employed to assess whether variables followed a normal distribution. In both visual and auditory tasks, we utilized the Wilcoxon rank sum test (also known as the Mann-Whitney U test) to evaluate the effects of pulse sequences. The error bars represent the standard error of the mean (SEM). Significance thresholds were set at $^*$ $p < 0.05$ and $^{**}$ $p < 0.01$, respectively.

## Results

Behavioral data was collected by conducting two speeded face-vs-car categorization experiments, one in auditory mode and one in visual mode. Participants were required to categorize noisy images, in visual task, or noisy sounds in auditory task in trials with one or two 50 ms pulses (Fig 1A and 1B). The single-pulse and double-pulse trials were randomly mixed together. Moreover, the strength of each pulse, image/sound, and the temporal gap between pulses changed randomly across trials. In double pulse trials, pulses always had a consistent object, a face or a car. Before data collection, subjects were extensively trained to achieve high levels of accuracy and were made aware of this consistency (refer to Materials and Methods).

### Integrating information from visual and auditory object cues over time

To analyze the behavioural performance of our subjects in visual and auditory tasks in single-pulse trials, we fitted two separate Weibull functions (Eq 6) to the data obtained from visual and auditory single-pulse trials (Fig 2A and 2B). We computed the corresponding threshold ($\alpha v = 0.30$, $\alpha_A = 0.45$) and slope ($\beta_v = 3.16$, $\beta_A = 1.37$) parameters for these psychometric functions by pooling the data collected from 15 subjects in each task. In the auditory task, the stimulus strength is equal to the proportion of signal with respect to the summation of signal and noise; for instance, 50% stimulus strength means one part signal and one part noise. Our results indicated that participants could enhance their accuracy by increasing the strength of

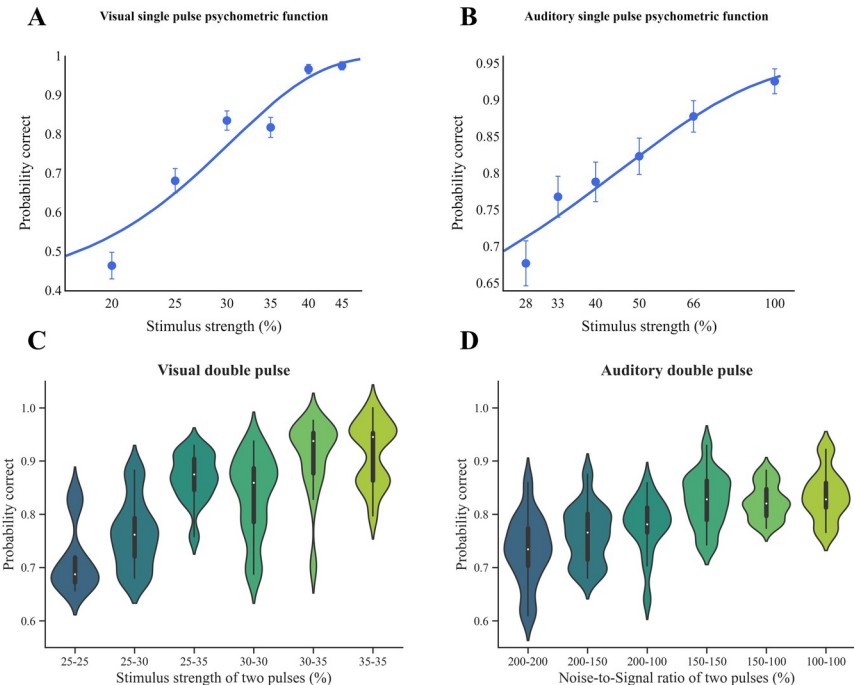

**Fig 2. Accuracy increases with stimulus strength in both single and double-pulse trials for visual and auditory tasks.** (A) Accuracy in single pulse trials in the visual task. (B) Accuracy in single pulse trials in the auditory task. In both figures, the horizontal axis displays the stimulus strength of single pulse trials and the vertical axis shows the subjects' performance as probability correct. The curve is the fit of a Weibull psychometric function (Eq 6) to the pooled sensory data across all subjects. Single pulse performance, whether visual or auditory, was improved by increasing the strength of stimuli. Error bars indicate SEM. (C) Accuracy in double-pulse trials in the visual task. The x-axis shows the stimulus strength of two pulses. Increasing the strength of the visual stimulus of two pulses improved the perceptual accuracy of participants. (D) Accuracy in double pulse trials in the auditory task. The horizontal axis shows the noise-to-signal ratio of two pulses. The evidence became stronger as the noise-to-signal ratio decreased. The upward trend was also seen in the performance of participants as the strength of the auditory stimuli increased in double -pulse trials. (C, D) The violon plots are based on the performance of 15 subjects in each condition for both sensory modalities. We calculated the probability of correct in double-pulse trials with unequal pulse strength by pooling data across two possible orders.

stimuli in visual and auditory tasks. In double-pulse trials, the participant's performance was improved when the total evidence received from two pulses increased (Fig 2C and 2D). In other words, participants were able to enhance their accuracy by aggregating more information from both pulses over time, regardless of the sensory modality of the stimuli.

## Stronger stimuli in double pulse trials improve subjects' sensitivity in discriminating between face and car in visual and auditory tasks

From the subjects' responses in double pulse trials, it is possible to calculate the d-prime value, $d'$, which reflects their sensitivity in discriminating between two noisy signals. Specifically, we determined $d'$ by comparing the means of the first signal (face) and the second signal (car) distributions, with a higher value indicating greater sensitivity. Our findings revealed an increase in the $d'$ value as the aggregated strength of stimuli of double pulse trials increased in both visual and auditory modes, indicating improved discrimination ability.

Moreover, the $d'$ value was higher in double-pulse trials than in single-pulse trials, and increasing the strength of each pulse while keeping the other pulse constant led to a corresponding increase in the $d'$ value in both sensory modalities (Fig 3A and 3B).

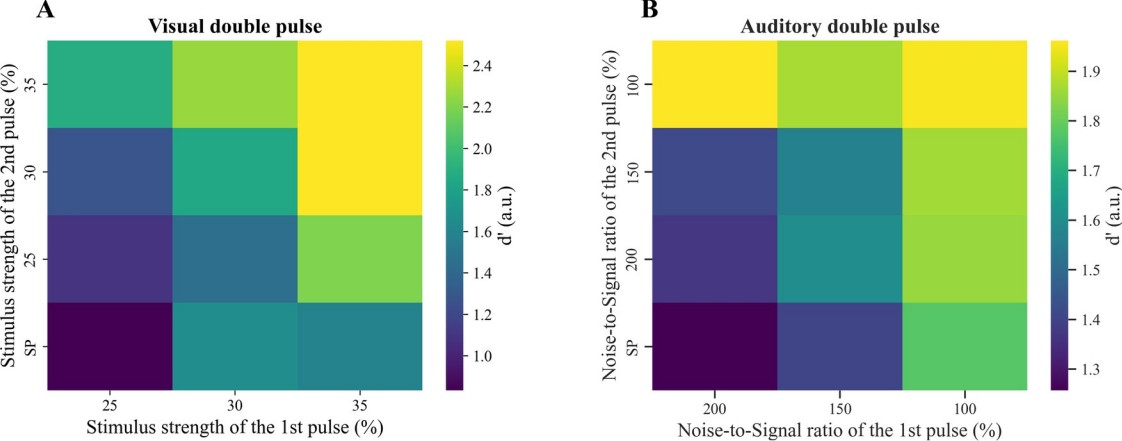

**Fig 3. Participants' sensory discrimination improves with increased combined stimulus strength in both visual and auditory tasks.** (A) D-prime value in single pulse and double pulse trials in the visual task. The x-axis shows the stimulus strength of the first pulse. The y-axis shows the stimulus strength of the second pulse. (B) D-prime value in single pulse and double pulse trials in the auditory task. The horizontal axis shows the noise-to-signal ratio of the first pulse. The vertical axis shows the noise-to-signal ratio of the second pulse. (A, B) In both figures, participants' ability to discriminate between face and car in both visual and auditory tasks improved with an increase in the aggregated stimuli strength of two pulses. The tables enable us to compare all potential trial comparisons in a specific sensory modality task. In both tasks, the d-prime values were calculated by pooling data from all subjects in each condition.

## The same decision policy is applied to the temporal gap between two pulses in both sensory modalities

In our study, we utilized a brief stimulus presentation of 50 ms in each pulse, shorter than the 120 ms presentation used in the double pulse motion direction discrimination task conducted by Kiani et al. [32]. However, it is important to note that the gap duration between pulses varied between 0 ms and 1080 ms, similar to the experimental design employed by Kiani et al. [32]. By selecting sub-second inter-pulse intervals, we ensured our study remained focused on temporal dynamics relevant to perceptual decision-making processes. This choice allowed us to investigate how the brain integrates discrete sensory evidence over short timescales, essential for understanding perceptual decision-making in many real-world conditions. The previous findings demonstrated that performance could be modelled as a linear integration of sensory evidence (more precisely, logistic regression), independent of the interval between cues. Building upon this solution and decision policy, we explored how varying gap durations between pulses impact the decision-making process in visual and auditory object categorization tasks.

We modelled our two data sets using linear integration (logistic regression) (Eq 2). This model could be considered a solution for the decision-making process in our different sensory experiments. Our findings revealed that accuracy in double-pulse trials was independent of inter-pulse interval in both visual and auditory modalities, with no significant difference observed across different gap durations between pulses (Fig 4A and 4B).

Specifically, the temporal gaps between pulses did not significantly impact accuracy in visual double-pulse trials (Fig 4A) with equal pulse strength (Eq 2, $p > 0.01$ for $\beta_3, \beta_4$; Table 1) and unequal pulse strength (Eq 2, $p > 0.01$ for $\beta_3, \beta_4, \beta_5$; Table 1), as well as in auditory double-pulse trials with equal pulse strength (Eq 2, $p > 0.01$ for $\beta_3, \beta_4$; Table 1) and unequal pulse strength (Eq 2, $p > 0.01$ for $\beta_3, \beta_4, \beta_5$; Table 1). So, the evidence from the first pulse remains

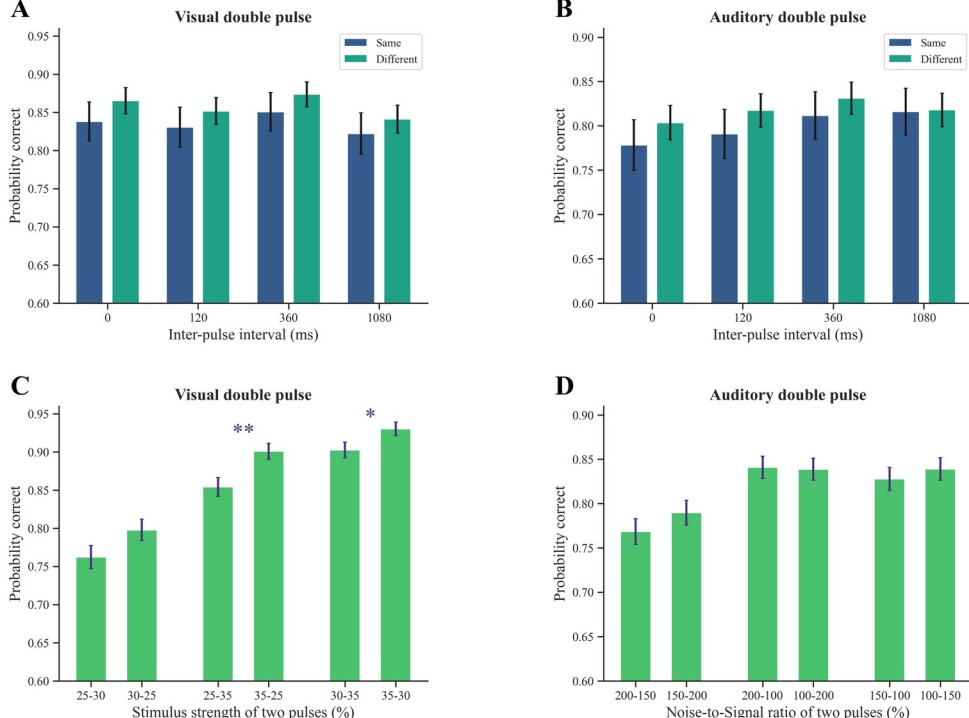

**Fig 4. A shared decision policy is applied to the temporal gap, while distinct policies are applied to the sequence of pulses in visual and auditory tasks.** (A, B) Accuracy in visual and auditory double-pulse trials in different temporal gaps. In each sensory modality, double-pulse trials were divided into two parts: trials with pulses of the same strength and trials with pulses of different strengths. In visual mode, no significant difference was observed in participants' performance across double-pulse trials with the same stimuli in different temporal gaps ($F_{3,8} = 0.03$ $p = 0.99$). This finding was also consistent with the results of double-pulse trials with different pulses in the visual modality ($F_{3,8} = 0.11$ $p = 0.95$), as well as double-pulse trials with the same auditory stimuli and different auditory stimuli ($F_{3,8} = 0.26$ $p = 0.85$, $F_{3,8} = 0.25$ $p = 0.86$). In both tasks, one-way ANOVA was used and probability correct for the same/different double-pulse trials was calculated by pooling data from all subjects. (C) Accuracy depended on the sequence of visual pulses. (D) Accuracy did not depend on the sequence of auditory pulses. In the visual task, the strong-weak pulse sequence resulted in higher accuracy than the weak-strong sequence, while in the auditory task, both pulses were equally important. In both tasks, probability correct for double-pulse trials with unequal pulse strength was calculated by pooling data across all intertrial intervals. We employed the Wilcoxon rank sum test, also known as the Mann-Whitney U test, to assess the effects of pulse sequences in both visual and auditory tasks. The differences in pulse sequence strength between unequal visual double pulses and their inverses were as follows: 25–30 and 30–25 (p-value = 0.08), 25–35 and 35–25 (p-value = 0.004), and 30–35 and 35–30 (p-value = 0.04), respectively. However, no statistically significant differences were observed between unequal auditory double pulses and their inverses (Fig 4C, 4D). Error bars indicate SEM. * $p < 0.05$, ** $p < 0.01$.

relatively stable throughout the gap duration between two pulses, indicating the robustness of keeping information up to 1 s between pulses in the brain for both sensory modalities.

## Different decision policies are applied to the sequence of pulses in visual and auditory tasks

In a perceptual decision-making process based on gathering two pulses of information over time, it is unclear whether the brain assigns equal importance to each pulse or exhibits a preference for one over the other. This ambiguity also exists in natural stimulus discrimination tasks that involve solely visual or auditory information. Previous research on double pulse motion direction discrimination tasks [32] showed that the choice accuracy is dependent on the sequence of motion pulses. Specifically, the weak-strong pulse sequence results in higher

**Table 1. Performance was largely unaffected by inter-pulse interval for double-pulse trials with equal pulse strength and with unequal pulse strength.**

| Equations | States | $\beta_1$ | $\beta_2$ | $\beta_3$ | $\beta_4$ | $\beta_5$ |
|---|---|---|---|---|---|---|
| Eq 2 | Visual: Different | 0.54 ± 0.06 ($p \cong 0$) | 0.48 ± 0.06 ($p \cong 0$) | 0.42 ± 0.57 ($p = 0.46$) | 0.16 ± 0.34 ($p = 0.64$) | -0.64 ± 0.33 ($p = 0.06$) |
| | Auditory: Different | 0.15 ± 0.05 ($p \cong 0$) | 0.15 ± 0.05 ($p \cong 0$) | -0.31 ± 0.4 ($p = 0.44$) | 0.16 ± 0.24 ($p = 0.49$) | 0.17 ± 0.24 ($p = 0.46$) |
| | Visual: Same | 0.65 ± 0.07 ($p \cong 0$) | | 0.04 ± 0.39 ($p = 0.9$) | -0.09 ± 0.41 ($p = 0.81$) | |
| | Auditory: Same | 0.16 ± 0.06 ($p \cong 0$) | | -0.55 ± 0.28 ($p = 0.06$) | 0.65 ± 0.3 ($p = 0.03$) | |
| Eq 3 | Visual | 0.47 ± 0.04 ($p \cong 0$) | -0.13 ± 0.03 ($p \cong 0$) | | | |
| | Auditory | 0.18 ± 0.03 ($p \cong 0$) | -0.002 ± 0.03 ($p = 0.96$) | | | |
| Eq 4 | Visual | 0.85 ± 0.23 ($p \cong 0$) | 0.69 ± 0.23 ($p \cong 0$) | -0.58 ± 0.34 ($p = 0.09$) | | |
| | Auditory | 0.88 ± 0.30 ($p \cong 0$) | 0.88 ± 0.30 ($p \cong 0$) | -0.71 ± 0.30 ($p = 0.01$) | | |

Each row shows the coefficients of different equations (mean ± SE) and their associated *p* values.

accuracy when compared to the strong-weak sequence. In other words, the decision-making process is more affected by the strength of the second pulse. However, it remains unclear whether this phenomenon extends to other similar tasks with different stimuli.

**The first pulse is more important than the second in the visual task.** Our findings indicate that subjects were more accurate when a strong pulse was presented before a weak pulse, compared with the opposite sequence in the visual task. This result is further supported by Fig 4C, which illustrates that participants achieved higher accuracy in double-pulse trials involving unequal pulse strength when the stronger motion stimulus appeared first (Eq 3, $\beta_2 =$ -0.013 ± 0.03, $p < 0.001$). Additionally, as Table 1 shows, the value of $\beta_1$ is higher than $\beta_2$ in Eqs 2 and Eq 4 which shows that the subjects were more sensitive to the first pulse than the second pulse. Finally, this increased efficacy was not dependent on the strength of the second pulse (Eq 4, $\beta_3 =$ -0.58 ± 0.34, $p > 0.05$).

**Equal importance of the first and second pulse in the auditory task.** In the auditory mode, our results are different and it seems that the brain used a different approach to process information for its final decisions (Fig 4D). Surprisingly, both brief pulses of information had the same value for the brain, and the choice accuracy did not depend on the sequence of pulses (Eq 3, $\beta_2 =$ -0.002 ± 0.03, $p = 0.96$). Moreover, Table 1 shows that the values of $\beta_1$ and $\beta_2$ in Eqs 2 and 4 are equal, which implies that the subjects are equally sensitive to both pulses. Notably, there were also potential interactions between the two audio pulses (Eq 4, $\beta_3 =$ -0.71 ± 0.30, $p = 0.01$).

## Confidence is in line with accuracy in both sensory modalities

The concept of confidence is linked to the sense of assurance in one's decisions, judgments, or ideas [36, 37]. For instance, in our previous example, if an observer perceives two obscure images or sounds with a temporal gap and subsequently decides to flee the scene, to what extent does he believe his decision is correct? This inner feeling can even affect his escape speed and his future decisions.

To study confidence, we asked participants to rate their confidence after their responses in both visual and auditory tasks (Fig 1A and 1B). They had to choose among six different-sized bars at the end of each trial. The smallest bar meant no confidence (zero) and the largest bar meant full confidence (one). The other bars had confidence values of 0.2, 0.4, 0.6, and 0.8, respectively. We considered that the first three bars correspond to low confidence and the last three correspond to high confidence.

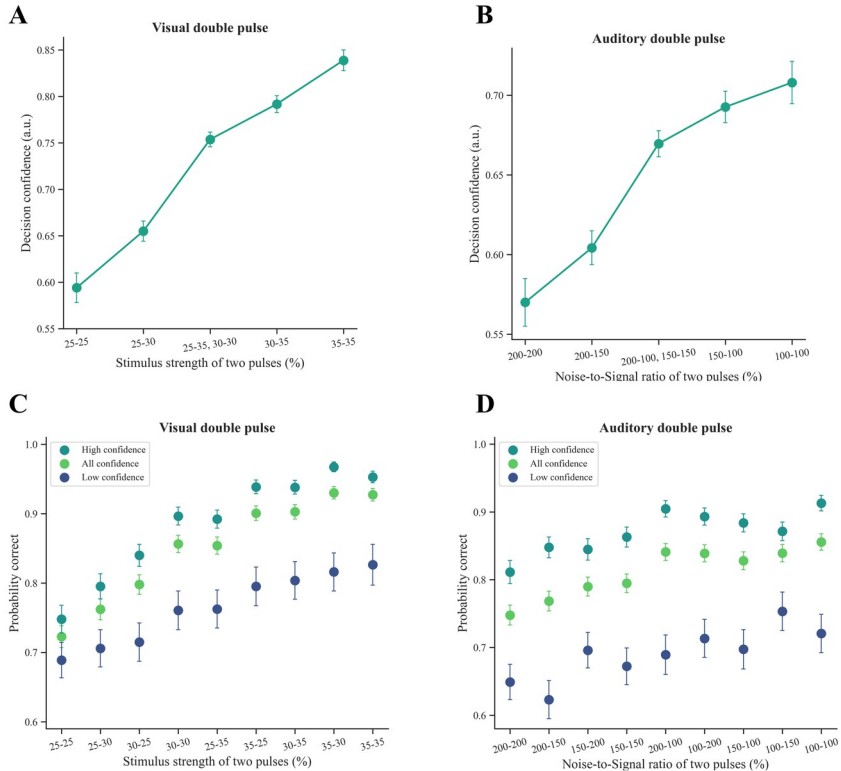

**Fig 5. Participants' confidence increases with aggregated stimulus strength and correlates with accuracy in both visual and auditory tasks.** (A, B) Subjects' confidence in the visual and auditory tasks. In both tasks, participants' confidence improved as the aggregated stimuli strength of two pulses increased. (C, D) The relationship between self-reported confidence and performance in visual and auditory tasks. In all double pulse trials and in both sensory modalities, subjects' performance was higher when they reported high confidence than when they reported low confidence. Error bars indicate SEM.

We examined how the confidence or the perceived probability of being correct was related to the strength of stimuli in visual and auditory double pulse trials (Fig 5). We found that as the sum of the strength of the two visual or auditory pulses increased, the participants' confidence improved (Fig 5A and 5B). Moreover, in both tasks, participants were more accurate when they reported higher confidence in all double pulse trials. In other words, subjects' performance was better when they had high confidence compared to when they had low confidence and vice versa (Fig 5C and 5D), which means that the confidence accords with the performance of subjects.

## Discussion

One of the common types of perceptual decision-making is decisions based on the accumulation of evidence that is presented separately over time. In real-world decisions, with exposure to naturalistic stimuli, this evidence can be perceived with different sensory modalities. So, in this study, we designed an experiment with two distinct modes: visual and auditory. In each mode, subjects were shown two pulses of information about the same object, face or car, often separated by a time interval.

Our main results demonstrated that the accuracy of double-pulse trials is unaffected by the temporal gap of up to one second between the two stimuli in both visual and auditory modes.

Moreover, we indicated that while both pulses are equally important in the auditory mode, the first pulse holds greater significance than the second one in the visual mode. The results of this study, along with a previous study [32], reveal that in perceptual decision-making tasks that involve gathering distinct evidence over time and with fixed action, the decision solution could be the same (e.g., linear integration), while the decision policy could either remain the same (e.g., choice accuracy in double-pulse trials is independent of the inter-pulse interval) or change flexibly (e.g., which pulse has the larger effect on decisions) according to the context. Meanwhile, the participants' performance remains relatively similar across all tasks.

Here, we observed participants could improve their accuracy effectively by integrating information from two pulses in both visual and auditory tasks. In many studies, d-prime which is one of the most common measures of sensitivity in signal detection theory, has been used to quantify an observer's ability to discriminate between signal and noise or two signals including object categorization tasks [50, 51]. Our findings demonstrated increasing the aggregated strength of visual and auditory stimuli in double-pulse trials increased the value of d-prime. This growth in d-prime value indicates an improvement in participants' ability to recognize objects (faces vs cars). More specifically, we observed similar results in both sensory modalities when the strength of the first pulse was kept constant and the strength of the second pulse was increased, or when the strength of the second pulse was kept constant and the strength of the first pulse was increased. So, the brain effectively uses the visual or auditory information of both pulses to improve its choice accuracy, which is similar to the findings of previous studies with different visual stimuli [5, 32].

Crucially in this work, we were able to study flexibility in choosing decision policies. In all decisions, decision-makers must determine the context [52, 53], set up the solution, decision policies, and associate sensory inputs to actions [4]. In this work, it was shown that linear integration is the appropriate solution for both visual and auditory tasks. Additionally, both tasks have the same action. Therefore, it would be possible to study flexibility in decision policy. Initially, we observed the same policy in both visual and auditory tasks. The accuracy of double-pulse trials was independent of the temporal gap between the two stimuli. This finding aligns with the effect of the inter-pulse interval on the double-pulse motion direction discrimination task in prior research by Kiani et al [32]. This result shows that the brain can retain and integrate sensory information effectively over short temporal gaps (up to one second) without loss of accuracy, highlighting robust neural mechanisms for maintaining sensory evidence. This ability also supports adaptive decision-making when sensory information is temporarily absent for varying short durations. Moreover, although previous research [32, 34] has shown that the sequence of visual motion pulses can affect choice accuracy, we cannot extend this phenomenon to the auditory object categorization task. In the auditory task, both pulses hold equal importance, while in the visual mode, the first pulse is more important than the second, which is in contrast to the prior result with random dot motion stimulus [32]. Thus, the human brain can flexibly change its decision policies when facing different contexts, which helps it achieve optimal performance in any task.

The differential effect of pulse sequencing observed between visual and auditory tasks can be attributed to modality-specific processing pathways. In visual tasks, visual stimuli often have enduring representations, and the first pulse might influence subsequent processing. Moreover, the initial visual stimulus may capture attention more strongly, leading to better encoding and a greater impact on decision-making, while later stimuli may receive less attention or be interpreted relative to the first [54]. Conversely, auditory processing does not show this primacy effect, likely due to its high temporal resolution and capability for continuous integration, which facilitates equal weighting of sequential inputs [55, 56].

Previous studies shed light on the neural mechanisms that enable the brain to efficiently integrate sensory information across temporal gaps. It was shown that persistent neural activity in regions such as the posterior parietal cortex allows sensory representations to be maintained during delays, supporting the integration process [57]. Additionally, working memory processes, particularly in the dorsolateral prefrontal and posterior parietal cortices, facilitate temporary storage of information for effective combination across intervals [58, 59]. Expanding on these mechanisms, Azizi and Ebrahimpour (2023) [33] propose a model that emphasizes the interplay of recurrent cortical dynamics within the centro-parietal and frontal brain areas. This dynamic interaction facilitates the efficient maintenance and integration of sensory evidence over time, ensuring accurate and cohesive decision-making.

We also studied the relationship between accuracy and confidence by investigating the same object categorization task in two different sensory contexts. Our study revealed that when participants reported low confidence levels in each sensory modality and at each level of strength of stimuli, their accuracy was lower compared to when they reported high confidence levels. Furthermore, in both tasks, as the strength of stimuli increased, the subjects' decision confidence also increased. These results align with previous research [39, 41, 60] and emphasize the importance of participants' confidence level in determining their performance, regardless of the sensory modality of inputs.

As we illustrated, sensory input is a factor that could influence the brain's flexibility in the decision-making process. Living in a dynamic environment leads us to make decisions based on the information we gather through various sensory modalities. These are then processed via different pathways in the brain. For instance, in the visual pathway, visual information is transmitted from the retina via the optic nerve to the visual cortex. At this point, intricate processing is carried out to recognize objects, shapes, and patterns. However, the auditory pathway begins with the ear picking up sound waves. These waves are subsequently transformed into neural signals and sent to the auditory cortex, where pitch, volume, and location are interpreted. Therefore, studying how the brain integrates information from different senses to make decisions and guide behavior is important. Research has shown that both rats and humans can combine auditory and visual information to improve the accuracy of their choices [19]. Additionally, brain activity reflects better discriminability of motion direction and correlates with the perceptual benefit provided by congruent multisensory information [20]. Recent studies have also demonstrated that complementary audiovisual information can enhance our ability to make decisions compared to visual information alone [18]. Furthermore, it was demonstrated that multisensory integration and decision-making processes are continuously interconnected [21]. In multisensory decision-making studies, auditory and visual stimuli are usually presented simultaneously to participants. However, in this study, we examined the integration of distinct visual information over time and distinct auditory information over time separately. This approach allowed us to investigate the integration of information in the visual and auditory pathways individually and provided a foundation for studying multisensory decision-making based on gathering discrete information over time in future.

## Conclusion

To understand the flexibility of perceptual decision-making in different contexts, we need to examine the generalizability of previous research to a wider range of stimuli, including naturalistic stimuli that can be perceived through different sensory modalities. In this study, we conducted two almost identical experiments through visual and auditory sensory modalities that are similar to a comparable experiment in a previous study [32]. We found that participants

choose the same decision solutions in all tasks but adjust their decision policies flexibly depending on the context.

## Limitations and future work

To enhance statistical power and generalizability, future studies should include more participants and avoid author involvement to reduce potential bias. Furthermore, it would be valuable to investigate the impact of other sensory modalities, such as olfactory and haptic senses, on the perceptual decision-making process. This exploration could provide further insights into how different sensory modalities influence the brain's flexibility in selecting decision policies in multisensory environments. Longer temporal gap durations could also be considered to encompass a broader range of conditions in future research.

## Acknowledgments

We express our gratitude to E. Karimzadeh and A. Khatami, for their invaluable cooperation in data acquisition. We also extend our appreciation to all participants involved in our experiments.

## Author Contributions

**Conceptualization:** Masoumeh Golmohamadian, Mehrbod Faraji, Fatemeh Fallah, Fatemeh Sharifizadeh, Reza Ebrahimpour.

**Data curation:** Masoumeh Golmohamadian, Mehrbod Faraji.

**Formal analysis:** Masoumeh Golmohamadian, Mehrbod Faraji, Fatemeh Fallah, Fatemeh Sharifizadeh.

**Funding acquisition:** Masoumeh Golmohamadian, Fatemeh Fallah.

**Investigation:** Masoumeh Golmohamadian, Mehrbod Faraji, Fatemeh Fallah, Fatemeh Sharifizadeh, Reza Ebrahimpour.

**Methodology:** Masoumeh Golmohamadian, Mehrbod Faraji, Fatemeh Fallah, Fatemeh Sharifizadeh, Reza Ebrahimpour.

**Project administration:** Reza Ebrahimpour.

**Resources:** Masoumeh Golmohamadian, Mehrbod Faraji, Fatemeh Fallah, Fatemeh Sharifizadeh, Reza Ebrahimpour.

**Software:** Masoumeh Golmohamadian, Mehrbod Faraji, Fatemeh Fallah, Fatemeh Sharifizadeh.

**Supervision:** Masoumeh Golmohamadian, Reza Ebrahimpour.

**Validation:** Masoumeh Golmohamadian, Mehrbod Faraji, Fatemeh Fallah, Fatemeh Sharifizadeh, Reza Ebrahimpour.

**Visualization:** Masoumeh Golmohamadian, Mehrbod Faraji, Fatemeh Fallah, Fatemeh Sharifizadeh.

**Writing – original draft:** Masoumeh Golmohamadian.

**Writing – review & editing:** Masoumeh Golmohamadian, Mehrbod Faraji, Fatemeh Fallah, Reza Ebrahimpour.

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
