## [Decision Letter · Decision Letter 0]

25 Oct 2024

PONE-D-24-31715Flexibility in Choosing Decision Policies in Gathering Discrete Evidence Over TimePLOS ONE

Dear Dr. Ebrahimpour,

Thank you for submitting your manuscript to PLOS ONE. After careful consideration, we feel that it has merit but does not fully meet PLOS ONE’s publication criteria as it currently stands. Therefore, we invite you to submit a revised version of the manuscript that addresses the points raised during the review process. Please submit your revised manuscript by Dec 09 2024 11:59PM. If you will need more time than this to complete your revisions, please reply to this message or contact the journal office at plosone@plos.org. Please include the following items when submitting your revised manuscript:A rebuttal letter that responds to each point raised by the academic editor and reviewer(s). You should upload this letter as a separate file labeled 'Response to Reviewers'.A marked-up copy of your manuscript that highlights changes made to the original version. You should upload this as a separate file labeled 'Revised Manuscript with Track Changes'.An unmarked version of your revised paper without tracked changes. You should upload this as a separate file labeled 'Manuscript'.If applicable, we recommend that you deposit your laboratory protocols in protocols.io to enhance the reproducibility of your results. Protocols.io assigns your protocol its own identifier (DOI) so that it can be cited independently in the future. For instructions see: https://journals.plos.org/plosone/s/submission-guidelines#loc-laboratory-protocols. Additionally, PLOS ONE offers an option for publishing peer-reviewed Lab Protocol articles, which describe protocols hosted on protocols.io. Read more information on sharing protocols at https://plos.org/protocols?utm_medium=editorial-email&utm_source=authorletters&utm_campaign=protocols.

We look forward to receiving your revised manuscript.

Kind regards,

Alexandra Kavushansky, PhD

Academic Editor

PLOS ONE

Journal Requirements:

file:///C:/Users/e754871/Desktop/PRTC/PRTC%202024/October%202024/October%2025,%202024/PONE-D-24-31715/Journal%20requirements:%20%0b%0bWhen%20submitting%20your%20revision,%20we%20need%20you%20to%20address%20these%20additional%20requirements.%20%0b%0b1.%20Please%20ensure%20that%20your%20manuscript%20meets%20PLOS%20ONE's%20style%20requirements,%20including%20those%20for%20file%20naming.%20The%20PLOS%20ONE%20style%20templates%20can%20be%20found%20at%20https:/journals.plos.org/plosone/s/file%3fid=wjVg/PLOSOne_formatting_sample_main_body.pdf%20and\ When submitting your revision, we need you to address these additional requirements.

Reviewers' comments:

Reviewer's Responses to Questions

**Comments to the Author**

1. Is the manuscript technically sound, and do the data support the conclusions?

Reviewer #1: Yes

2. Has the statistical analysis been performed appropriately and rigorously? 

Reviewer #1: I Don't Know

3. Have the authors made all data underlying the findings in their manuscript fully available?

Reviewer #1: No

4. Is the manuscript presented in an intelligible fashion and written in standard English?

Reviewer #1: Yes

5. Review Comments to the Author

Reviewer #1: The study “Flexibility in Choosing Decision Policies in Gathering Discrete Evidence Over Time“ offers insights into the brain’s adaptive strategies in decision-making across different sensory modalities. It highlights the brain's ability to maintain a consistent method of integrating evidence while flexibly adjusting how that evidence is weighted based on the context provided by the sensory modality. This underscores the brain's flexibility in optimizing decision-making processes in diverse environments.

By using the same object categorization task (faces vs. cars) and assuming linear integration as the decision solution, the authors attribute any differences in performance to changes in decision policies.

The primary discovery of this study is that while the brain utilizes a consistent decision solution (linear integration of evidence) across different sensory modalities (visual and auditory), it flexibly adapts its decision policies based on the sensory context. Specifically:

- Visual Tasks: The brain places greater weight on the first pulse of evidence, making the initial stimulus more influential in decision-making.

- Auditory Tasks: The brain treats both pulses of evidence equally, showing no preference for the sequence in which auditory information is received.

The authors conclude that the brain employs a consistent method (decision solution) of integrating evidence over time across different sensory modalities but adapts its decision policies (strategies for weighting evidence) based on the context provided by the sensory modality. This flexibility allows the brain to optimize decision-making processes in dynamic environments and suggests that cognitive strategies are not fixed but can be tailored to the demands of the situation.

Overall, the manuscript presents a meaningful contribution to the field of cognitive neuroscience and perceptual decision-making by demonstrating the brain’s flexibility in decision policies across sensory modalities. While the core findings are compelling, addressing the limitations mentioned below would strengthen the manuscript.

Main comments:

1. Provide clearer definitions of key terms, particularly decision policies and decision solutions, early in the manuscript.

2. Elaborate on the implications of the findings and potential neural mechanisms involved. e.g.,

What is the interpretation of a lack of effect for inter-pulse interval?

or,

Why in visual, but not auditory, does the sequence of pulse strength matter?

Also please justify temporal gap selection: why were sub-second intervals chosen, and not larger inter pulse intervals?

3. Sample size (15 subjects with 2 being authors) is relatively small. Please consider adding more subjects, or acknowledge the sample size limitations and potential biases, suggesting ways future research can address these issues.

4. To make it easier for the reader to skim the figures and understand the paper, please change all figure titles to complete sentences that convey the main takeaway of the figure; e.g. Figure 3 title should be like:

"Participants' sensory discrimination improves with increased combined stimulus strength in both visual and auditory tasks"

Minor comments:

1. Remove the extra "and" in figure 3 title. (made bold below):

Figure 3. Participants' sensitivity (d-prime) in single and double-pulse trials and in both visual and auditory tasks

2. Figure 4 title: remove the dash after decision

6. PLOS authors have the option to publish the peer review history of their article (what does this mean?). If published, this will include your full peer review and any attached files.

Reviewer #1: No

---

## [Author Response · Author response to Decision Letter 0]

19 Nov 2024

Review Comments to the Author

Reviewer #1: The study “Flexibility in Choosing Decision Policies in Gathering Discrete Evidence Over Time“ offers insights into the brain’s adaptive strategies in decision-making across different sensory modalities. It highlights the brain's ability to maintain a consistent method of integrating evidence while flexibly adjusting how that evidence is weighted based on the context provided by the sensory modality. This underscores the brain's flexibility in optimizing decision-making processes in diverse environments.

By using the same object categorization task (faces vs. cars) and assuming linear integration as the decision solution, the authors attribute any differences in performance to changes in decision policies.

The primary discovery of this study is that while the brain utilizes a consistent decision solution (linear integration of evidence) across different sensory modalities (visual and auditory), it flexibly adapts its decision policies based on the sensory context. Specifically:

- Visual Tasks: The brain places greater weight on the first pulse of evidence, making the initial stimulus more influential in decision-making.

- Auditory Tasks: The brain treats both pulses of evidence equally, showing no preference for the sequence in which auditory information is received.

The authors conclude that the brain employs a consistent method (decision solution) of integrating evidence over time across different sensory modalities but adapts its decision policies (strategies for weighting evidence) based on the context provided by the sensory modality. This flexibility allows the brain to optimize decision-making processes in dynamic environments and suggests that cognitive strategies are not fixed but can be tailored to the demands of the situation.

Overall, the manuscript presents a meaningful contribution to the field of cognitive neuroscience and perceptual decision-making by demonstrating the brain’s flexibility in decision policies across sensory modalities. While the core findings are compelling, addressing the limitations mentioned below would strengthen the manuscript.

Main comments:

1: Provide clearer definitions of key terms, particularly decision policies and decision solutions, early in the manuscript.

Authors: Thank you for your suggestion to clarify key terms. Below, we present more precise definitions of "decision policies" and "decision solutions," supported by a broader range of examples with references.

Flexibility in decision-making can be categorized into three levels: flexibility in task solutions, flexibility in decision policies, and flexibility in stimulus-action mapping (Okazawa & Kiani, 2023). At the highest level, decision-makers can adopt different solutions for a task, such as integrating, differentiating, or even disregarding sensory inputs depending on the context. For example, evidence accumulation is the optimal solution for tasks involving the discrimination or categorization of stable sensory information (Waskom, & Kiani, 2018; Stine et al., 2013). However, other task conditions may call for different approaches. When comparing the magnitudes of two stimuli, subtracting inputs is more effective than integrating them (Romo & Lafuente, 2013; Machens et al., 2005), whereas, for detection tasks against a stable background, differentiation is a better strategy than integration. At the next level, decision-makers adjust decision policies by modifying parameters within decision-making models to achieve optimal outcomes. These adjustments support flexible computation, enabling decision-makers to weigh sensory inputs, prioritize evidence, assess costs and rewards (Doi et al., 2020), and manage urgency to respond (Forstmann et al., 2008). Finally, flexibility in stimulus-action mapping allows decision-makers to associate sensory inputs with appropriate actions, emphasizing the adaptable nature of decision-making processes.

These definitions have been incorporated into the relevant paragraph in the introduction. Below is the revised version of that paragraph.

“In the literature, flexibility in decision-making, often referred to as context dependency, reflects the brain's ability to adjust its behavior and neural activity across different tasks to meet varying demands. This adaptability is organized into three levels, each representing distinct types of flexibility that enhance decision making performance (Okazawa & Kiani, 2023). At the highest level, decision-makers can adopt different solutions for a task, such as integrating, differentiating, or even disregarding sensory inputs depending on the context. For example, evidence accumulation is the optimal solution for tasks involving the discrimination or categorization of stable sensory information (Waskom, & Kiani, 2018; Stine et al., 2013). However, other task conditions may call for different approaches. When comparing the magnitudes of two stimuli, subtracting inputs is more effective than integrating them (Romo & Lafuente, 2013; Machens et al., 2005), whereas, for detection tasks against a stable background, differentiation is a better strategy than integration. At the next level, decision-makers adjust decision policies by modifying parameters within decision-making models to achieve optimal outcomes. These adjustments support flexible computation, enabling decision-makers to weigh sensory inputs, prioritize evidence, assess costs and rewards (Doi et al., 2020), and manage urgency to respond (Forstmann et al., 2008). Finally, perceptual decisions involve mapping sensory stimuli to actions, but existing decision-making models typically assume a fixed stimulus-action association and focus on evidence integration rather than flexible stimulus-action mapping. Together, these levels of flexibility demonstrate that decision-making is not simply about processing information but involves a multi-layered, adaptive system capable of dynamically adjusting solutions and policies in response to the immediate context, thereby enabling more effective navigation of complex and real situations.”

Okazawa, G., & Kiani, R. (2023). Neural mechanisms that make perceptual decisions flexible. Annual review of physiology, 85(1), 191-215.

Waskom, M. L., & Kiani, R. (2018). Decision making through integration of sensory evidence at prolonged timescales. Current Biology, 28(23), 3850-3856

Stine, G. M., Zylberberg, A., Ditterich, J., & Shadlen, M. N. (2020). Differentiating between integration and non-integration strategies in perceptual decision making. Elife, 9, e55365.

Romo, R., & de Lafuente, V. (2013). Conversion of sensory signals into perceptual decisions. Progress in neurobiology, 103, 41-75.

Machens, C. K., Romo, R., & Brody, C. D. (2005). Flexible control of mutual inhibition: a neural model of two-interval discrimination. Science, 307(5712), 1121-1124.

Doi, T., Fan, Y., Gold, J. I., & Ding, L. (2020). The caudate nucleus contributes causally to decisions that balance reward and uncertain visual information. Elife, 9, e56694.

Forstmann, B. U., Dutilh, G., Brown, S., Neumann, J., Von Cramon, D. Y., Ridderinkhof, K. R., & Wagenmakers, E. J. (2008). Striatum and pre-SMA facilitate decision-making under time pressure. Proceedings of the National Academy of Sciences, 105(45), 17538-17542.

2- Elaborate on the implications of the findings and potential neural mechanisms involved. e.g., What is the interpretation of a lack of effect for inter-pulse interval?

or, Why in visual, but not auditory, does the sequence of pulse strength matter?

Also please justify temporal gap selection: why were sub-second intervals chosen, and not larger inter pulse intervals?

Authors: We appreciate the reviewer's insightful comments and the opportunity to elaborate on our findings and their implications. Below, we address each of the points raised, including additional relevant literature to strengthen our explanations.

a) What is the interpretation of a lack of effect for inter-pulse interval?

Authors: The lack of an effect of inter-pulse interval on decision accuracy in both visual and auditory tasks suggests that the brain is capable of maintaining and integrating sensory evidence over temporal gaps of up to approximately one second without significant decay. This finding underscores the robustness of the neural mechanisms responsible for preserving sensory information from the initial pulse across sub-second intervals. Such robustness is crucial for effective decision-making in a dynamic environment and enables the brain to adapt to the irregular timing of sensory signals typically encountered in everyday life. Moreover, this finding aligns with previous research demonstrating that humans can integrate sensory evidence over temporal gaps without substantial loss of information (Kiani et al., 2013; Waskom & Kiani, 2018; Tohidi-Moghaddam et al., 2019).

To explain the observed lack of effect for the inter-pulse interval, we have added the green text in the following paragraph of the manuscript’s discussion section.

“Crucially in this work, we were able to study flexibility in choosing decision policies. In all decisions, decision-makers must determine the context, set up the solution, decision policies, and associate sensory inputs to actions. In this work, it was shown that linear integration is the appropriate solution for both visual and auditory tasks. Additionally, both tasks have the same action. Therefore, it would be possible to study flexibility in decision policy. Initially, we observed the same policy in both visual and auditory tasks. The accuracy of double-pulse trials was independent of the temporal gap between the two stimuli. This finding aligns with the effect of the inter-pulse interval on the double-pulse motion direction discrimination task in prior research by Kiani et al., 2013. This result shows that the brain can retain and integrate sensory information effectively over short temporal gaps (up to one second) without loss of accuracy, highlighting robust neural mechanisms for maintaining sensory evidence. This ability also supports adaptive decision-making when sensory information is temporarily absent for varying short durations. Moreover, although previous research has shown that the sequence of visual motion pulses can affect choice accuracy, we cannot extend this phenomenon to the auditory object categorization task. In the auditory task, both pulses hold equal importance, while in the visual mode, the first pulse is more important than the second, which is in contrast to the prior result with random dot motion stimulus. Thus, the human brain can flexibly change its decision policies when facing different contexts, which helps it achieve optimal performance in any task.”

Potential Neural Mechanisms:

The maintenance of sensory evidence may invol ve persistent activity in neural circuits, particularly within prefrontal or parietal cortices. Leon and Shadlen (2003) demonstrated that neurons in the posterior parietal cortex can sustain representations of sensory information over delays, facilitating integration processes. Such sustained neural activity could keep the representation of the first pulse active during the inter-pulse interval, allowing it to be effectively integrated with the second pulse upon its arrival. Moreover, the integration over temporal gaps likely engages working memory mechanisms. The dorsolateral prefrontal cortex and posterior parietal cortex are known to support the temporary storage and manipulation of sensory information, which could underlie the observed integration across intervals (Gold & Shadlen, 2007; Curtis & D'Esposito, 2003). Recent research by Azizi and Ebrahimpour (2023) provides valuable insights into the neural mechanisms involved in integrating sensory evidence presented with temporal gaps. They demonstrated that decisions based on the accumulation of evidence from separated cues over time are best explained by the interplay of recurrent cortical dynamics of centro-parietal and frontal brain areas. This interplay enables the brain to maintain and combine sensory information over delays efficiently.

Based on the above points, we have added the following paragraph to the discussion section and inserted the necessary references:

“Previous studies shed light on the neural mechanisms that enable the brain to efficiently integrate sensory information across temporal gaps. It was shown that persistent neural activity in regions such as the posterior parietal cortex allows sensory representations to be maintained during delays, supporting the integration process (Leon & Shadlen, 2003). Additionally, working memory processes, particularly in the dorsolateral prefrontal and posterior parietal cortices, facilitate temporary storage of information for effective combination across intervals (Gold & Shadlen, 2007; Curtis & D’Esposito, 2003. Expanding on these mechanisms, Azizi and Ebrahimpour (2023) propose a model that emphasizes the interplay of recurrent cortical dynamics within the centro-parietal and frontal brain areas. This dynamic interaction facilitates the efficient maintenance and integration of sensory evidence over time, ensuring accurate and cohesive decision-making.”

Kiani, R., Churchland, A. K., & Shadlen, M. N. (2013). Integration of direction cues is invariant to the temporal gap between them. Journal of Neuroscience, 33(42), 16483-16489.

Waskom, M. L., & Kiani, R. (2018). Decision making through integration of sensory evidence at prolonged timescales. Current Biology, 28(23), 3850-3856.

Tohidi-Moghaddam, M., Zabbah, S., Olianezhad, F., & Ebrahimpour, R. (2019). Sequence-dependent sensitivity explains the accuracy of decisions when cues are separated with a gap. Attention, Perception, & Psychophysics, 81, 2745-2754.

Leon, M. I., & Shadlen, M. N. (2003). Representation of time by neurons in the posterior parietal cortex of the macaque. Neuron, 38(2), 317-327.

Gold, J. I., & Shadlen, M. N. (2007). The neural basis of decision making. Annu. Rev. Neurosci., 30(1), 535-574.

Curtis, C. E., & D'Esposito, M. (2003). Persistent activity in the prefrontal cortex during working memory. Trends in cognitive sciences, 7(9), 415-423.

Azizi, Z., & Ebrahimpour, R. (2023). Explaining Integration of Evidence Separated by Temporal Gaps with Frontoparietal Circuit Models. Neuroscience, 509, 74-95.

b) Why in visual, but not auditory, does the sequence of pulse strength matter?

Authors: The differential effect of pulse sequence in the visual and auditory tasks likely reflects modality-specific processing characteristics in the brain. Our finding that pulse sequence impacts visual but not auditory processing can be explained by modality-specific neural pathways and the mechanisms of temporal integration. The visual system is known for its sensitivity to sequential patterns and is adept at processing time-dependent information in a structured way. This sensitivity is associated with cortical regions such as the visual cortex, where temporal and spatial dynamics are more finely tuned to detect changes and sequences. Auditory pathways, in contrast, often employ a rapid, more continuous processing mode optimized for different types of temporal coding (e.g., integration over rapid sound sequences without much variation)(Okazawa and Kiani, 2023). 

Visual Task:

In the visual modality, we observed that the first pulse had a greater influence on the decision than the second pulse. This may be due to visual stimuli often having enduring representations and may also capture attention more effectively, leading to enhanced encoding and greater influence on the decision-making process (Carrasco, 2011). Subsequent stimuli might be subject to reduced attentional resources or might be interpreted in the context of the initial percept. Finally, sustained activity in visual cortical areas could preferentially maintain the information from the first pulse. The ventral visual stream, involved in object recognition, may prioritize initial inputs, affecting the integration process (Di Lollo et al., 2000).

Auditory Task:

In contrast, the auditory system appears to weigh both pulses equally. Auditory information is inherently temporal and transient, and the auditory system specializes in processing 

---

## [Editor Report · Decision Letter 1]

10 Dec 2024

Flexibility in choosing decision policies in gathering discrete evidence over time

PONE-D-24-31715R1

Dear Dr. Ebrahimpour,

We’re pleased to inform you that your manuscript has been judged scientifically suitable for publication and will be formally accepted for publication once it meets all outstanding technical requirements.

Kind regards,

Alexandra Kavushansky, PhD

Academic Editor

PLOS ONE
---

## [Editor Report · Acceptance letter]

3 Jan 2025

PONE-D-24-31715R1 

PLOS ONE

Dear Dr. Ebrahimpour, 

I'm pleased to inform you that your manuscript has been deemed suitable for publication in PLOS ONE. Congratulations! Your manuscript is now being handed over to our production team.

Kind regards, 

on behalf of

Dr. Alexandra Kavushansky 

Academic Editor

PLOS ONE